# Challenges and Learning Curves in Adopting TaTME and Robotic Surgery for Rectal Cancer: A Cusum Analysis

**DOI:** 10.3390/cancers14205089

**Published:** 2022-10-18

**Authors:** Pere Planellas, Lídia Cornejo, Anna Pigem, Núria Gómez-Romeu, David Julià-Bergkvist, Nuria Ortega, José Ignacio Rodríguez-Hermosa, Ramon Farrés

**Affiliations:** 1Colorectal Surgery Unit, Department of General and Digestive Surgery, University Hospital of Girona, 17007 Girona, Spain; 2Girona Biomedical Research Institute (IDIBGI), 17007 Girona, Spain

**Keywords:** rectal cancer, taTME, robotic, learning curves

## Abstract

**Simple Summary:**

Rectal cancer surgery remains a challenge and information about the learning curve in adopting new techniques is lacking. This paper analyzes our experience in taTME (since 2015) and robotic surgery (since 2018) at a fully accredited referral center for the treatment of rectal cancer in Spain. In this retrospective study, we aim to analyze the learning curves for taTME and robot-assisted rectal procedures in the incorporation of these platforms into our practice. We sought to describe our team’s experience in incorporating these techniques and to analyze the difficulties that we have had. Hoping that sharing our experience can help other groups improve their results during the difficult initial phase of incorporating new techniques.

**Abstract:**

New techniques are being developed to improve the results of laparoscopic surgery for rectal cancer. This paper analyzes the learning curves for transanal total mesorectal excision (taTME) and robot-assisted surgery in our colorectal surgery department. We analyzed retrospectively data from patients undergoing curative and elective surgery for rectal cancer ≤12 cm from the anal verge. We excluded extended surgeries. We used cumulative sum (CUSUM) curve analysis to identify inflection points. Between 2015 and 2021, 588 patients underwent surgery for rectal cancer at our center: 67 taTME and 79 robot-assisted surgeries. To overcome the operative time learning curve, 14 cases were needed for taTME and 53 for robot-assisted surgery. The morbidity rate started to decrease after the 17th case in taTME and after the 49th case in robot-assisted surgery, but it is much less abrupt in robot-assisted group. During the initial learning phase, the rate of anastomotic leakage was higher in taTME (35.7% vs. 5.7%). Two Urological lesions occurred in taTME but not in robot-assisted surgery. The conversion rate was higher in robot-assisted surgery (1.5% vs. 10.1%). Incorporating new techniques is complex and entails a transition period. In our experience, taTME involved a higher rate of serious complications than robot-assisted surgery during initial learning period but required a shorter learning curve.

## 1. Introduction

Rapidly evolving technology has driven improvements in surgical techniques. Minimally invasive laparoscopic surgery has long been the standard for most operations. However, in rectal cancer, the non-inferiority of laparoscopic surgery over open surgery was not established until 2015 [1,2]. In parallel to the implementation of laparoscopic surgery for rectal cancer, other minimally invasive techniques were developed to overcome technical limitations of the laparoscopic for total mesorectal excision (TME). Robot-assisted surgery first appeared in 2006 [3], and transanal total mesorectal excision (taTME) first appeared in 2009 [4]. These techniques aim to improve the results of rectal cancer surgery in intraoperative measures (conversion and surgical time), pathology measures (quality of the mesorectum, circumferential, and distal margins), and postoperative outcomes (leakage, rate of definitive ostomies, quality of life, functional quality, survival, and local recurrence). However, whether one of these techniques is superior to the others remains to be demonstrated [5].

The steps involved in adopting new techniques vary widely among teams. Scientific societies have warned of the risks of incorporating new techniques without proper prior training [6].

Rectal cancer surgery remains a challenge for surgeons, and limited quality of evidence is available about the learning curve for adopting new techniques. It is difficult to compare the learning curves from previous analyses, which usually focus on the results of a single expert surgeon with previous experience in multiple techniques and abundant resources for support. Proponents of taTME and robot-assisted surgery have argued that although these techniques are more expensive than conventional laparoscopic surgery, they can achieve better surgical outcomes and require less time to learn [7]. The reported learning curve for applying these techniques in rectal cancer ranges from 15 to 43 cases for robot-assisted surgery [8,9,10] and from 40 to 71 cases for taTME [11,12,13,14]. The learning curve of laparoscopic rectal resections is approximately 16–35 cases [15,16,17].

In this paper, we describe our team learning curve since the incorporation of taTME in 2015 and Da Vinci in 2018 in the treatment of rectal cancer in our colorectal surgery department. Initially, taTME was indicated for middle or lower rectal cancer in cases considered complex (male sex, narrow pelvis, and/or bulky tumor) [18]. However, since the incorporation of robot-assisted surgery at our center the criteria for taTME has become more restrictive over time. Our goal was never to directly compare these two techniques; rather, we sought to describe our team’s experience in incorporating these techniques and to analyze the difficulties that we have had.

## 2. Materials and Methods

### 2.1. Study Design and Patient Selection

A retrospective analysis was conducted of a prospective cohort of all consecutive rectal resections who underwent elective curative taTME or robot-assisted surgery for rectal cancer situated ≤12 cm from the anal verge at our center between January 2015 and September 2021. We excluded patients who required extended surgeries (abdominoperineal resection, pelvic exenteration, and/or lateral pelvic lymphadenectomy).

### 2.2. Ethical and Quality Considerations

The institutional ethics committee approved the study (N° 2021.177), and patients provided written informed consent for the inclusion. In reporting the trial, we have followed the guidelines specified in the STROBE (Strengthening The Reporting of Observational Studies in Epidemiology) statement [19].

### 2.3. Preoperative Staging, Treatment, and Management

All patients underwent complete colonoscopies, and all diagnoses were histologically confirmed in colonoscopy specimens. Rectal tumors were staged by pelvic magnetic resonance imaging and/or rectal ultrasound. All patients also underwent chest and abdominopelvic computed tomography.

Patients with locally advanced rectal tumors (cT3/cT4 and/or cN1/cN2) underwent neoadjuvant long-course radiation therapy based on fractioned radiation (1.8 Gy per day; total dose, 45–50.4 Gy) with concomitant 5-fluracil (5-FU) or capecitabine; the response to neoadjuvant treatment was evaluated 8 to 10 weeks after completion of the regimen.

Perioperative management was uniform for all patients throughout the study period, including preoperative antibiotic prophylaxis and Enhanced Recovery After Surgery (ERAS) management [20].

### 2.4. Training and Surgical Technique

All surgeries analyzed were performed by surgeons from our colorectal surgery unit with extensive experience in laparoscopic surgery for rectal cancer. Before adopting taTME, our surgeons attended technical courses including the visualization of live surgeries and online videos as well as practice in a simulator and in cadavers. To introduce the technique, we established a stable team of two surgeons, who progressively alternated in the roles of lead and assistant surgeon.

In adopting robot-assisted surgery, we followed the Intuitive-Abex^®^ training program: self-learning, simulated model practice, and proctorship during the first three cases. Other surgeons were gradually and progressively incorporated following the training program.

In all taTME operations, we used a standardized synchronous double-team technique in which two surgical teams operated simultaneously. An abdominal team used laparoscopy to perform a medial-to-lateral mobilization, high tie of the inferior mesenteric artery, high tie of the inferior mesenteric vein at the lower border of the pancreas, complete or near-complete mobilization of the splenic flexure and total mesorectal excision. A transanal team used a single-port device inserted through the anus to place purse-string sutures distal to the tumor, rectotomy, and ascending dissection in the mesorectal plane. Air seal technology (AirSeal System Surgiquest, Milford, CT, USA) was incorporated In October 2016.

For robot-assisted surgery, we used the Da Vinci System (Intuitive Surgical, Sunnyvale, CA, USA), using the Da Vinci Si robot for the first four cases and the Da Vinci Xi robot for the remaining cases.

### 2.5. Data Collection

We analyzed the following demographic and clinical variables: sex; body mass index (BMI); age; distance of tumor from the anal verge; physical status according to the American Society of Anesthesiologists (ASA) score, dichotomized into I or II/III or IV; and neoadjuvant treatment.

We analyzed the following intra- and post-operative variables: surgical technique, conversion to open surgery, operative time, blood loss, anastomosis and/or stoma creation, postoperative morbidity, major postoperative complications, anastomotic leakage, and length of stay.

We analyzed the following pathology variables: distal and circumferential margin involvement (<1 mm), mesorectum quality, and number of lymph nodes harvested.

### 2.6. Study Outcomes

The primary outcome was to describe the number of operations required to decrease the postoperative morbidity rate in patients undergoing taTME and in those undergoing robot-assisted surgery.

Secondary outcomes were to report the number of operations required to decrease the mean operative time, major complications rate, and anastomotic leakage rate in the two approaches.

Tertiary outcomes were the conversion rates and pathology outcomes in the two approaches.

### 2.7. Endpoints

The endpoints for evaluating the learning curve were postoperative morbidity; operative time, defined as the time from incision to the closing of the last wound; major postoperative complications, defined as grade III or higher on the Clavien–Dindo classification [21]; and anastomotic leakage, defined as early or delayed leak, pelvic abscess, anastomotic fistula, chronic sinus, or anastomotic stricture [22].

### 2.8. Cumulative Sum Analysis

Cumulative sum analysis (CUSUM) is often used to evaluate the learning curve for surgical techniques [23]. For cumulative sum analyses, cases were arranged chronologically on the *X*-axis from the earliest to the most recent. The only continuous variable analyzed with CUSUM was operative time. The cumulative sum of operative time (CUSUMOTn) expresses the difference between the operative time for the case and the mean operative time for all cases; it was calculated using the formula CUSUMOTn = Σni (xi − μ), where I is the number of the case, μ is the mean operative time for the entire series, and xi is the operative time for the case.

The categorical variables analyzed with CUMSUM were postoperative morbidity, major postoperative complications, anastomotic leakage, and surgical failure. We used the standard non-risk-adjusted CUSUM chart to analyze these variables. If the event occurred, the case was scored as [—(1—the risk of the series)]; if the event did not occur, the case was scored as [—(0—the risk of the series)]. The cumulative sum of scores after each case was then calculated in chronological order.

### 2.9. Statistical Analysis

Categorical variables are summarized as absolute and relative frequencies. To compare categorical variables between groups, we used chi-square tests or Fisher’s exact test, as appropriate. Continuous variables are summarized as means and standard deviations (SD) or medians and interquartile ranges (IQR), as appropriate. We used the Kolmogorov–Smirnov test to determine whether distributions were normal. To compare continuous variables between groups, we used Student’s t-test or the Mann–Whitney test as appropriate for comparisons between two groups and ANOVA for comparisons between three groups. Statistical significance was set at *p* < 0.05. We used SPSS v. 20.0 (SPSS Inc., Chicago, IL, USA) for all analyses.

## 3. Results

### 3.1. Patients

Between January 2015 and September 2021, 588 patients underwent elective curative surgery for rectal cancer at our hospital. Of these, 189 (32.1%) patients underwent taTME (*n* = 67) or robot-assisted surgery (*n* = 122). We excluded 22 patients whose tumors were located > 12 cm from the anal verge and 21 patients who required extended surgery. Thus, we analyzed data from 146 patients, 67 who underwent taTME and 79 who underwent robot-assisted surgery (Figure 1).

Two surgeons performed all taTME (TS1, 53 surgeries; and TS2, 14 surgeries). Four surgeons performed all robot-assisted surgeries (RS1, 29 surgeries; RS2, 28 surgeries; RS3, 13 surgeries; and RS4, 9 surgeries).

Table 1 reports patients’ baseline characteristics. Of those patients who underwent taTME surgery a 97.0% were men, median value of BMI was 28 kg/m^2^, and a 73.1% of tumors were located under 7 cm from anal verge.

In total, 63.3% of patients who underwent robot-assisted surgery were men, median value of BMI was 26 kg/m^2^, and the 31.6% of tumors were located under 7 cm from anal verge.

### 3.2. Surgical and Pathology Outcomes

Table 2 reports the surgical and pathology variables for the two surgical approaches. A greater proportion of ultralow anterior resections and of diverting ileostomies were reported in the taTME approach (68.7% and 89.6%, respectively). However, the proportion of anastomoses was 80–90% in both approaches. Similar bleeding and operative time were observed in taTME and robotic-assisted surgery.

The conversion rate was higher in robot-assisted surgery (10.1%). The rates of postoperative morbidity were around 30% in both surgical approaches. Major complication rates were observed in 6.3% of patients who underwent robotic surgery and in 11.9% of those who underwent the taTME approach.

In total, 14.9% of patients who underwent taTME surgery and 5.1% who underwent robotic-assisted surgery had anastomotic leakages. Incomplete mesorectum was observed in 13.4% of patients who underwent taTME and in 6.3% of those who underwent robotic surgery. Similar rates of distal and circumferential margin involvement quality were observed in both approaches.

### 3.3. Learning Curve Analyses through CUSUM Control Charts

#### 3.3.1. Postoperative Morbidity

In taTME, the postoperative morbidity rate was 34.3%. The slope of the CUSUM curve for postoperative complications in taTME became positive after the 19th case, indicating a decrease in postoperative complications (Figure 2A). Postoperative morbidity was significantly lower after this inflection point (68.4% cases 1–19 vs. 20.8% cases 20–67, *p* < 0.001).

In robot-assisted surgery, the postoperative morbidity rate was 31.6%. The CUSUM curve for postoperative complications in robot-assisted surgery was more complex. It reached a low point after 49 cases, but there were no significant differences in the rate of complications between the two phases delineated by this inflection point (34.7% before vs. 26.7% after, *p* = 0.457) (Figure 2B).

#### 3.3.2. Operative Time

In taTME, mean operative time was 245.2 min (IQR, 200–300 min), and three phases were observed. The first phase (cases 1–14) corresponds to the initial learning period. The second phase (cases 15–43), in which operative time decreased steadily and markedly to the minimum reached, corresponds to the consolidation period. The third phase (cases 44–67), in which more complex skills were acquired, corresponds to the mastery period. Mean operative time differed significantly among the three phases (250.5 ± 54.7 min in the first phase vs. 224 ± 51 min in the second phase vs. 267.9 ± 54.4 min in the third phase, *p* = 0.014) (Figure 2C). Table 3 reports the surgical outcomes for taTME during the three periods. In the mastery period, we observed significantly lower rates of bleeding (*p* = 0.029), postoperative morbidity (*p* = 0.016), and readmission (*p* = 0.013); moreover, unlike in the other periods, there were no cases of anastomotic leakage (*p* = 0.005) or major complications (*p* = 0.043). The results in the mastery phase were also influenced by a change in indications, reserving taTME only for more complex cases. Selecting only complex patients for taTME increased the operative time but did not worsen morbidity, leakages, or specimen quality because surgeons’ experience in the technique had already been consolidated.

In robot-assisted surgery, mean operative time was 247.1 min (IQR, 210–289 min), and two phases were observed. The first phase (cases 1–53) corresponds to the initial learning period, and the second phase (cases 54–79) corresponds to the consolidation/mastery period, in which operative time decreased, reaching a low point at the 75th case, and ascending thereafter (Figure 2D). Differences in mean operative between the two phases did not reach statistical significance (253.5 ± 56.2 min in the first phase vs. 234 ± 59.8 min in the second phase, *p* = 0.160). No significant differences between the two phases were observed in the rates of major complications, reinterventions, anastomotic leakage, or readmission (Table 4).

#### 3.3.3. Major Postoperative Complications

In taTME, the overall rate of major postoperative complications was 11.9%. The CUSUM curve for major postoperative complications had a negative slope until case 19, followed by a period of consolidation from case 19 to 36 and a positive slope thereafter, indicating that the necessary competence had been achieved. The rate of major complications differed significantly between the three phases identified (31.2% in the first phase vs. 11.8% in the second phase vs. 0% in the third phase, *p* = 0.001) (Figure 2E).

In robot-assisted surgery, the overall rate of major postoperative complication was 6.3%. The CUSUM curve for major postoperative complications had a positive slope until the 31st case, when it became negative; the cumulative sum increased from case 59 to 79, indicating a decrease in the number of major postoperative complications (Figure 2F). The difference in the rate of major complications between the two periods delineated by the inflection point did not reach statistical significance (8.6% for cases 1–58 vs. 0% for cases 59–79, *p* = 0.317).

#### 3.3.4. Anastomotic Leakage

In taTME, the overall rate of anastomotic leakage was 14.6%. The slope of the CUSUM curve was negative until the 19th case; after this inflection point, it became positive, indicating a decrease in the rate of anastomotic leakage. The difference in the rate of anastomotic leakage between the two periods delineated by the inflection point was significant (44.4% for cases 1–18 vs. 4.1% for cases 19–67, *p* < 0.001) (Figure 2G).

In robot-assisted surgery, the overall rate of anastomotic leakage was 5.1%. The slope of the CUSUM curve was positive until case 31, from which point it became negative until case 44, indicating that technical competence had been achieved. The difference in the rate of anastomotic leakage between the two periods delineated by the inflection point did not reach statistical significance (6.8% for cases 1–31 vs. 2.9% for cases 32–79, *p* = 0.625) (Figure 2H).

## 4. Discussion

We aimed to devise an overall outcome-based assessment of the adoption of new surgical techniques for the treatment of rectal cancer at our institution. To this end, we analyzed the team’s learning curves in incorporating taTME and robotic surgery in the treatment of rectal cancer, analyzing the difficulties that we have had.

In our experience, during the learning phase, the rate of anastomotic leakage and urological lesions was higher in taTME, but the rate of conversion was higher in robot-assisted surgery. The learning curves for operative time and decreased morbidity were shorter in taTME than in robot-assisted surgery. In our experience, the consolidation/mastery in TaTME includes phase I and II (44 cases). In comparison, robot-assisted surgery only includes phase I (53 cases).

It is difficult to compare the learning curves from our analyses with those in other reports, which usually focus on the results of a single expert surgeon with previous experience in multiple techniques and abundant resources for support. This approach tends to yield much more defined graphs that are easier to analyze. Undoubtedly, these reports provide valuable information. However, we considered it important to analyze the adoption of new techniques in our specialized colorectal surgery department where various surgeons work together in collaboration. Although this analytical approach is messier, it provides essential information that is impossible to obtain from focusing on a single expert surgeon.

Concerns have arisen about the safety and efficacy of taTME. Norway declared a moratorium on taTME after a new unexpected pattern of recurrence was observed early after surgery [24], and skeptics have recommended discontinuing its incorporation into new centers in some countries [25]. Learning taTME is challenging because it requires learning a new surgical field, often resulting in the appearance of previously uncommon complications (urethral injuries, CO2 embolism, iliac injury, etc.) and a risk of increasing the incidence of local recurrence. That is why a structured program for the incorporation of the technique has been established [6,26,27], and taTME is indicated only for cancers in the middle and lower thirds of the rectum in patients with unfavorable anatomical factors. Moreover, to prevent possible complications and difficulties during the learning curve, the technique is evolving to incorporate other approaches, such as transanal rectal transection with a singled-stapled anastomosis (TTSS) [28] and robotic-taTME [29]. Minimally invasive transanal surgery will continue to progress, and the indications might even be expanded because this approach confers various advantages. It facilitates access to local lesions and improves the treatment for anastomotic dehiscence [30]. The results of the randomized COLOR III and ETAP-GRECCAR 11 trials [31,32] should help clarify whether taTME conducted by expert teams provides benefits beyond those achieved with other techniques.

Robotic platforms are also evolving. More agile platforms incorporating navigation systems will likely become available at lower costs [33]. However, no formal, standard accreditation for robotic surgery exists [34]. One dominant manufacturer, Intuitive^®^, organizes regulated training and employs proctors to supervise the use of their systems during the initial phases of their implementation. Nevertheless, the acquisition of experience and skills in robotic surgery during residency remains unregulated, although scientific societies in many countries have created specific fellowships in robotic surgery separate and independent from colorectal surgery fellowships or other accreditations [35].

Many authors consider that the keys to the successful development of minimally invasive surgery include specialization, long training periods and the accumulation of experience, and ensuring that the most expert surgeons perform the operations [36]. Previous experience in laparoscopic surgery reduces the duration of the learning curve for other techniques [37,38].

It is imperative for training in new techniques to evolve. The Halsted approach characterized as ‘see one, do one, teach one’ must be replaced by more formal, structured learning. Simulators, cadaver training, and proctorship options are emerging [39]. As new techniques become more and more accessible without rigorous requirements, scientific societies will have to oversee the implementation of these techniques to ensure patient safety (Figure 3).

It is crucial for individual surgeons and surgical teams in accredited units to know their own results, so they can assess the need to change their strategy and determine whether to incorporate new techniques when they appear. The techniques and platforms we use today are likely to be replaced by others tomorrow.

### Limitations

Learning curve studies are complex and have many possible biases that compromise external validity. Our unit introduced taTME in 2015 and robot-assisted surgery in 2018, and we continue to use both techniques today. There are important differences between the patients who underwent taTME and those who underwent robot-assisted surgery. Patients undergoing taTME had lower lesions and were more likely to be men and to receive neoadjuvant treatment.

It is important to underline that we used different strategies in incorporating the two techniques into our colorectal surgery unit. When incorporating taTME, we decided to consolidate a stable transanal team of two surgeons. Realizing that this approach disrupted the balance of cases in the department, we decided to employ a different approach in incorporating robot-assisted surgery, opting for a progressive but accelerated incorporation of four surgeons into the robotic surgery team. All these surgeons had extensive prior experience in laparoscopic surgery.

We hope that sharing our experience can help other groups improve their results during the difficult initial phase of incorporating new techniques.

## 5. Conclusions

In our experience, the rate of anastomotic leakage and urological lesions during the initial learning phase was higher in taTME, but the conversion rate during this period was higher in robot-assisted surgery. The learning curves for operative time and decreased morbidity were shorter for taTME than for robot-assisted surgery. The learning phase is crucial during the incorporation of new techniques, and the risk of serious complications is high during this period, especially in taTME. Nevertheless, the risk involved in learning to apply new techniques must be counterbalanced against the risks and shortcomings of laparoscopic surgery; indeed, it is important to remember that the development of new techniques arises from the need to improve outcomes where established techniques present difficulties and complications.

It is essential to ensure optimal preparation before introducing new surgical techniques. Increased time practicing on experimental models, training, and proctoring programs have proven useful in the implementation of new techniques. These programs need to be definite and structured.

## Figures and Tables

**Figure 1 cancers-14-05089-f001:**
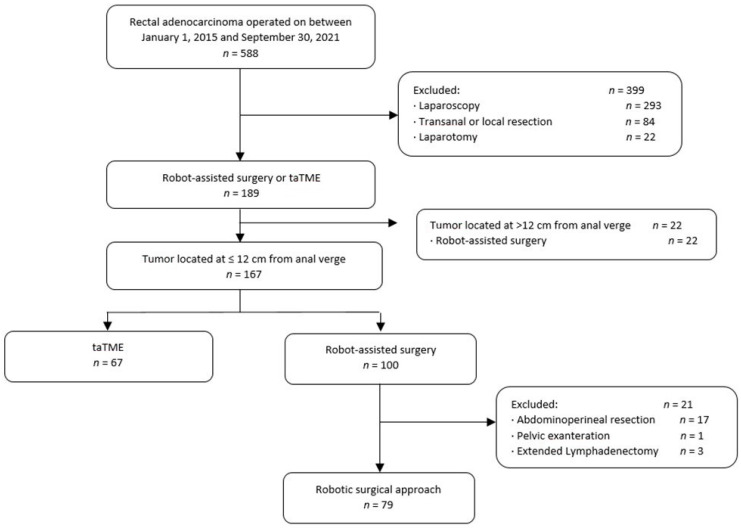
Flowchart of the study.

**Figure 2 cancers-14-05089-f002:**
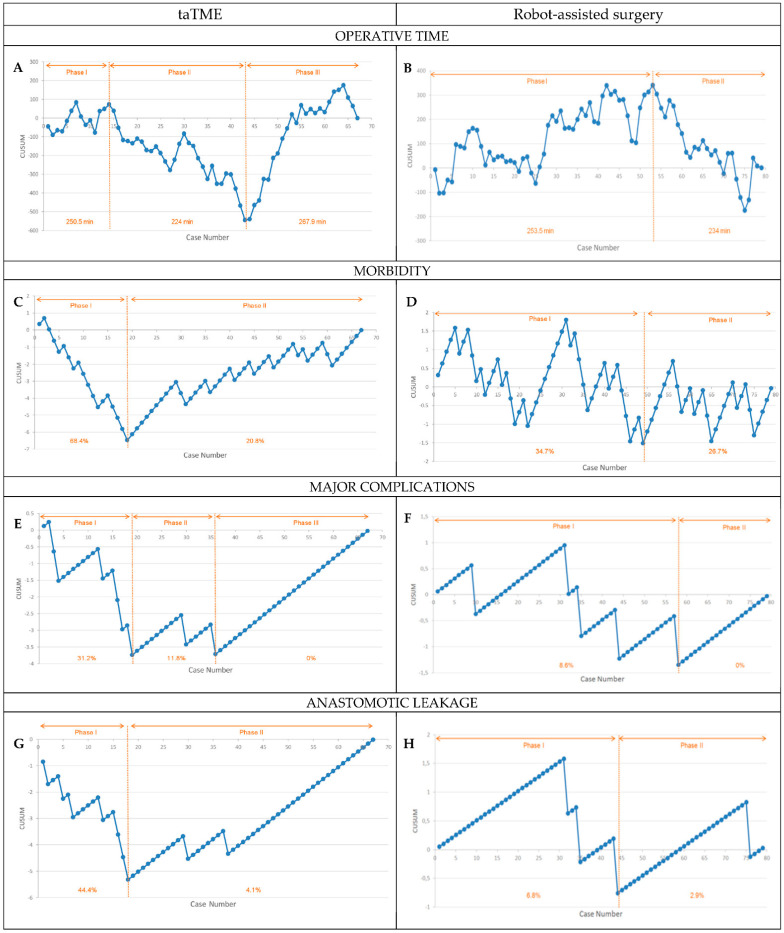
CUSUM learning curves for taTME and robot-assisted surgery. (**A**) taTME operative time; (**B**) Robot-assisted surgery operative time; (**C**) taTME morbidity; (**D**) Robot-assisted surgery morbidity; (**E**) taTME major complications; (**F**) Robot-assisted surgery major complications; (**G**) taTME anastomotic leakage; (**H**) Robot-assisted surgery anastomotic leakage.

**Figure 3 cancers-14-05089-f003:**
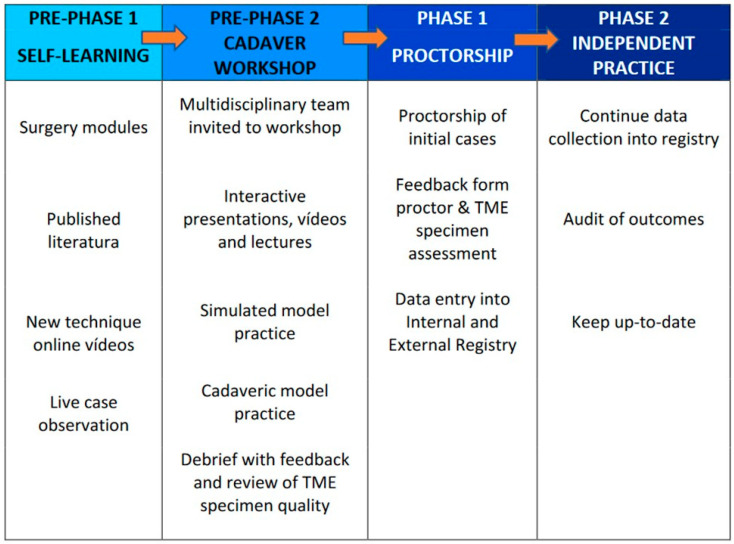
Training Curriculum for the Structured Adoption of New Techniques (Adapted from the Consensus on structured training curriculum for transanal total mesorectal excision (TaTME) [6]).

**Table 1 cancers-14-05089-t001:** Baseline characteristics of patients who underwent taTME or robot-assisted surgery for rectal cancer.

	Robot-Assisted Surgery *n* = 79	TaTME *n* = 67	Total *n* = 146	*p*-Value
Sex Women Men	29 (36.7%) 50 (63.3%)	2 (3.0%) 65 (97.0%)	31 (21.1%) 115 (78.8%)	<0.001
BMI, median (IQR)	26 (23–29)	28 (25–32)	26 (24–30)	0.217
Age, median (IQR)	70 (62–76)	64 (58–71)	67 (59–75)	0.008
Tumor location(distance from anal verge) ≤3 cm >3 cm and ≤7 cm >7 cm	1 (1.3%) 24 (30.3%) 54 (68.4%)	8 (11.9%) 41 (61.2%) 18 (26.9%)	9 (6.2%) 65 (44.5%) 72 (49.3%)	<0.001
ASA score ASA I or II ASA III or IV	6 (7.6%) 73 (92.4%)	10 (14.9%) 57 (85.1%)	16 (11.0%) 130 (89.0%)	0.158
cT stage T1 T2 T3 T4	5 (7.1%) 13 (18.6%) 42 (60.0%) 10 (14.3%)	0 (0%) 12 (19.0%) 48 (76.2%) 3 (4.8%)	5 (3.8%) 25 (18.8%) 90 (67.7%) 13 (9.8%)	0.026
cN stage N0 N1 N2	21 (30.4%) 25 (36.2%) 23 (33.3%)	9 (14.3%) 22 (34.9%) 32 (50.8%)	30 (22.7%) 47 (35.6%) 55 (41.7%)	0.045
Synchronous metastases No Yes	73 (92.4%) 6 (7.6%)	60 (89.6%) 7 (10.4%)	133 (91.1%) 13 (8.9%)	0.546
Neoadjuvant treatment No Yes	32 (40.5%) 47 (59.5%)	9 (13.4%) 58 (86.6%)	41 (28.1%) 105 (71.9%)	<0.001

BMI: Body Mass Index; ASA: American Society of Anesthesiologists; IQR = interquartile range.

**Table 2 cancers-14-05089-t002:** Surgical, postoperative, and oncological outcomes of patients who underwent robot-assisted surgery or taTME for rectal cancer.

	Robot-Assisted Surgery *n* = 79	TaTME *n* = 67	Total *n* = 146	*p*-Value
Surgical techniques Low anterior resection Ultralow anterior resection Hartmann	43 (54.4%) 23 (29.1%) 13 (16.5%)	14 (20.9%) 46 (68.7%) 7 (10.4%)	57 (39.0%) 69 (47.3%) 20 (13.7%)	<0.001
Anastomosis No Yes	13 (16.5%) 66 (83.5%)	7 (10.4%) 60 (89.6%)	20 (13.7%) 126 (86.3%)	0.293
Stoma No Stoma Colostomy Ileostomy	20 (25.3%) 13 (16.5%) 46 (58.2%)	0 (0%) 7 (10.4%) 60 (89.6%)	20 (13.7%) 20 (13.7%) 106 (72.6%)	<0.001
Estimated blood loss, in mL, median (IQR)	50 (50–50)	50 (50–100)	50 (50–50)	0.109
Operative time, in min, median (IQR)	240 (210–289)	240 (200–300)	240 (200–290)	0.868
Conversion to open surgery No Yes	71 (89.9%) 8 (10.1%)	66 (98.5%) 1 (1.5%)	137 (93.8%) 9 (6.2%)	0.039
Length of stay, in days, median (IQR)	6 (5–8)	6 (5–11)	6 (5–9)	0.039
Postoperative complications No Yes	54 (68.4%) 25 (31.6%)	44 (65.7%) 23 (34.3%)	99 (67.8%) 47 (32.2%)	0.611
Postoperative reintervention No Yes	75 (94.9%) 4 (5.1%)	59 (88.1%) 8 (11.9%)	134 (91.8%) 12 (8.2%)	0.145
Clavien-Dindo classification No complications <3A ≥3A	54 (68.4%) 20 (25.3%) 5 (6.4%)	44 (65.7%) 15 (22.3%) 8 (12.0%)	99 (67.8%) 35 (23.9%) 13 (8.3%)	0.484
Major complications No Yes	74 (93.7%) 5 (6.3%)	59 (88.1%) 8 (11.9%)	133 (91.1%) 13 (8.9%)	0.236
Anastomotic leakage No Yes	75 (94.9%) 4 (5.1%)	57 (85.1%) 10 (14.9%)	132 (90.4%) 14 (9.6%)	0.052
Rehospitalization No Yes	69 (87.3%) 10 (12.7%)	52 (77.6%) 15 (22.4%)	121 (82.9%) 25 (17.1%)	0.120
Mesorectal quality Complete/Partial Incomplete	74 (93.7%) 5 (6.3%)	58 (86.6%) 9 (13.4%)	132 (90.4%) 14 (9.6%)	0.146
Involvement of distal margin No Yes	75 (94.9%) 4 (5.1%)	65 (97.0%) 2 (3.0%)	140 (95.9%) 6 (4.1%)	0.688
Involvement of circumferential margin No Yes	74 (93.7%) 5 (6.3%)	65 (97.0%) 2 (3.0%)	139 (95.25) 7 (4.8%)	0.453

IQR = interquartile range.

**Table 3 cancers-14-05089-t003:** Comparison of operative time in different phases of the learning curve for taTME for rectal cancer.

	Phase I Learning *n* = 14	Phase II Competence *n* = 29	Phase III Mastery *n* = 24	Total *n* = 67	*p*-Value
Estimated blood loss, in mL, median (IQR)	50 (50–62.5)	50 (50–200)	50 (50–50)	50 (50–100)	0.029
Operative time, in min, median (IQR)	263.5 (200–292.5)	210 (180–257.5)	270 (224.3–315)	241 (200–297.5)	0.014
Conversion to open surgery No Yes	14 (100%) 0 (0%)	28 (96.6%) 1 (3.4%)	24 (100%) 0 (0%)	66 (98.5%) 1 (1.5%)	1.000
Length of stay, in days, median (IQR)	9 (5.8–22.8)	6 (5–10.5)	5.5 (5–7.8)	6 (5–11)	0.074
Postoperative complications No Yes	5 (35.7%) 9 (64.3%)	21 (72.4%) 8 (27.6%)	18 (75.0%) 6 (25.0%)	44 (65.7%) 23 (34.3%)	0.016
Clavien-Dindo classification No complications <3A ≥3A	5 (35.7%) 6 (42.9%) 3 (21.4%)	21 (72.4%) 3 (10.3%) 5 (17.2%)	18 (75.0%) 6 (25.0%) 0 (0%)	44 (65.7%) 15 (22.3%) 8 (12.0%)	0.009
Major complications No Yes	11 (78.6%) 3 (21.4%)	24 (82.8%) 5 (17.2%)	24 (100%) 0 (0%)	59 (88.1%) 8 (11.9%)	0.043
Postoperative reintervention No Yes	11 (78.6%) 3 (21.4%)	24 (82.8%) 5 (17.2%)	24 (100%) 0 (0%)	59 (88.1%) 8 (11.9%)	0.043
Anastomotic leakage No Yes	9 (64.3%) 5 (35.7%)	24 (82.8%) 5 (17.2%)	24 (100%) 0 (0%)	57 (85.1%) 10 (14.9%)	0.005
Rehospitalization No Yes	7 (50.0%) 7 (50.0%)	23 (79.3%) 6 (20.7%)	22 (91.7%) 2 (8.3%)	52 (77.6%) 15 (22.4%)	0.013

BMI: Body Mass Index; ASA: American Society of Anesthesiologists. IQR = interquartile range.

**Table 4 cancers-14-05089-t004:** Comparison of operative time in different phases of the learning curve for robot-assisted surgery for rectal cancer.

	Phase I Learning *n* = 53	Phase II Consolidation/Mastery *n* = 26	Total *n* = 79	*p*-Value
Estimated blood loss, in mL, median (IQR)	50 (50–50)	50 (50–50)	50 (50–50)	0.486
Operative time, in min, median (IQR)	250 (218–290)	217.5 (198.5–269.3)	240 (210–289)	0.160
Conversion to open surgery No Yes	49 (92.5%) 4 (7.5%)	22 (84.6%) 4 (15.4%)	71 (89.9%) 8 (10.1%)	0.428
Length of stay, in days, median (IQR)	7 (5–8.5)	5 (4–6.5)	6 (5–8)	0.143
Postoperative complications No Yes	36 (67.9%) 17 (32.1%)	18 (69.2%) 8 (30.8%)	54 (68.4%) 25 (31.6%)	1.000
Clavien-Dindo classification No complications <3A ≥3A	36 (67.9%) 13 (24.5%) 4 (7.5%)	18 (69.2%) 7 (26.9%) 1 (3.8%)	54 (68.4%) 20 (25.3%) 5 (6.4%)	1.000
Major complications No Yes	49 (92.5%) 4 (7.5%)	25 (96.2%) 1 (3.8%)	74 (93.7%) 5 (6.3%)	1.000
Postoperative reintervention No Yes	50 (94.3%) 3 (5.7%)	25 (96.2%) 1 (3.8%)	75 (94.9%) 4 (5.1%)	1.000
Anastomotic leakage No Yes	50 (94.3%) 3 (5.7%)	25 (96.2%) 1 (3.8%)	75 (94.9%) 4 (5.1%)	1.000
Rehospitalization No Yes	44 (83.0%) 9 (17.0%)	25 (96.2%) 1 (3.8%)	69 (87.3%) 10 (12.7%)	0.153

## Data Availability

The data presented in this study are available from the corresponding author upon request. The data are not publicly available due to ethical regulations.

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
