# Peer review of "Challenges and Learning Curves in Adopting TaTME and Robotic Surgery for Rectal Cancer: A Cusum Analysis"

_cancers, 2022, doi:10.3390/cancers14205089_

Round 1

Reviewer 1 Report

Thank you for the possibility to review the article entitled “Challenges and learning curves in adopting taTME and robotic 2 surgery for rectal cancer: a cusum analysis”. Overall, the manuscript is well written and the statistical analysis sound.

14 cases for learning TaTME is rather short, too short maybe. TaTME is an extremely difficult approach and I don’t believe 14 cases are enough. Maybe the phase II should be incorporated in the learning phase, but ok, I agree that your statistical analysis may be interpreted as you did.

Other biases such as sample size and different surgeons that approached TaTME and robotic (2 vs. 4) are correctly acknowledged.

I congratulate with the authors for this article, very interesting, well conducted and written and scientifically rigorous.

Author Response

Thank you for the possibility to review the article entitled “Challenges and learning curves in adopting taTME and robotic 2 surgery for rectal cancer: a cusum analysis”. Overall, the manuscript is well written and the statistical analysis sound.

Thank you for classifying the study as well-written and scientifically rigorous work.

14 cases for learning TaTME is rather short, too short maybe. TaTME is an extremely difficult approach and I don’t believe 14 cases are enough. Maybe the phase II should be incorporated in the learning phase, but ok, I agree that your statistical analysis may be interpreted as you did.

Thanks for pointing this out, as you mentioned phase I in TaTME seems to short. Phase I shows the initial difficulties in the incorporation of a new approach and modification route in the usual technique (especially in the technique to perform mechanical anastomosis). However it is not until we pass phase II that we reach the mastery phase (case 44).

Other biases such as sample size and different surgeons that approached TaTME and robotic (2 vs. 4) are correctly acknowledged. I congratulate with the authors for this article, very interesting, well conducted and written and scientifically rigorous.

Thank you for your comment.

Reviewer 2 Report

Dear Authors!

Thank you very much for giving me the chance to review the manuscript "Challenges and learning curves in adopting taTME and robotic surgery for rectal cancer: a cusum analysis" from Planellas et al. 

The manuscript contains important information about the learning curves of two different techniques for rectal resections with TME for mid and low rectal cancer. The manuscript style is adequately and the statistical analysis is very good. Some minor corrections could be done:

1. In the introduction section you describe that the intention never was to compare those techniques. However, this is not true. The whole manuscript is in a way a comparison of both. Especially morbidity(especially leakage rate) was higher in the taTME group - also in "phase 3". The reason for this might be that more complex cases were treated in the "more experienced phase 3". This should be more discussed in the discussion section. Furthermore, p-values in table 1 and table 2 would be desirable.  

2. Table 3: I think you meant Phase III (mastery) not Phase II.

Author Response

Thank you very much for giving me the chance to review the manuscript "Challenges and learning curves in adopting taTME and robotic surgery for rectal cancer: a cusum analysis" from Planellas et al. 

The manuscript contains important information about the learning curves of two different techniques for rectal resections with TME for mid and low rectal cancer. The manuscript style is adequately and the statistical analysis is very good.

Thank you for your comment.

Some minor corrections could be done:

1. In the introduction section you describe that the intention never was to compare those techniques. However, this is not true. The whole manuscript is in a way a comparison of both.

Thanks for pointing this out, the aim of the work has been to describe our experience during the learning curve of the two approaches, not the superiority of one over another. In order to determine which technique is better, we would have to carry out a study comparing outcomes once the mastery phase has reached in both approaches.

Especially morbidity (especially leakage rate) was higher in the taTME group - also in "phase 3". The reason for this might be that more complex cases were treated in the "more experienced phase 3". This should be more discussed in the discussion section.

Sorry that it was unclear, morbidity was around 30% in both approaches. It is true that anastomotic leakage rate was higher in TaTME. However, any case of anastomotic leakage was recorded in TaTME phase III.

We believe that the high rate of anastomotic leakage during the initial phases was due to the technical difficulty of performing a mechanical anastomosis in TaTME and the different machines used in the learning phase. But once we have learned the technique, we have drastically reduced the incidence of complications, despite selecting for this technique only the most unfavorable cases (men, obese, narrow pelvis).

Furthermore, p-values in table 1 and table 2 would be desirable.  

Thank you for the suggestion, p-values in table 1 and table 2 were included in the revised manuscript.

2. Table 3: I think you meant Phase III (mastery) not Phase II.

You are right; the mistake has been fixed in the revised manuscript.